# Artificial Neuronal Ensembles with Learned Context Dependent Gating

**Matthew J. Tilley, Michelle Miller & David J. Freedman**
Department of Neurobiology, University of Chicago
Chicago, IL 60637, USA
{mjtilley, mcmiller1, dfreedman}@uchicago.edu

## Abstract

Biological neural networks are capable of recruiting different sets of neurons to encode different memories. However, when training artificial neural networks on a set of tasks, typically, no mechanism is employed for selectively producing anything analogous to these neuronal ensembles. Further, artificial neural networks suffer from catastrophic forgetting, where the network's performance rapidly deteriorates as tasks are learned sequentially. By contrast, sequential learning is possible for a range of biological organisms. We introduce Learned Context Dependent Gating (LXDG), a method to flexibly allocate and recall 'artificial neuronal ensembles', using a particular network structure and a new set of regularization terms. Activities in the hidden layers of the network are modulated by gates, which are dynamically produced during training. The gates are outputs of networks themselves, trained with a sigmoid output activation. The regularization terms we have introduced correspond to properties exhibited by biological neuronal ensembles. The first term penalizes low gate sparsity, ensuring that only a specified fraction of the network is used. The second term ensures that previously learned gates are recalled when the network is presented with input from previously learned tasks. Finally, there is a regularization term responsible for ensuring that new tasks are encoded in gates that are as orthogonal as possible from previously used ones. We demonstrate the ability of this method to alleviate catastrophic forgetting on continual learning benchmarks. When the new regularization terms are included in the model along with Elastic Weight Consolidation (EWC) it achieves better performance on the benchmark 'permuted MNIST' than with EWC alone. The benchmark 'rotated MNIST' demonstrates how similar tasks recruit similar neurons to the artificial neuronal ensemble.

## 1 Introduction

### 1.1 Catastrophic Forgetting

Learning sequentially without forgetting prior tasks is commonly known as continual learning or life-long learning. When an artificial neural network is trained on a task and the same model is then trained on a new task, the model's performance on the initial task tends to drop significantly. The model tends to overwrite parameters that are important to prior learned tasks, leading to the well known problem referred to as 'catastrophic forgetting'. This phenomenon, initially coined as 'catastrophic interference', was first observed in 1989 (McCloskey & Cohen, 1989) and is a longstanding problem encountered when training artificial neural networks. This problem is related to the stability-plasticity dilemma (Mermillod et al., 2013). The dilemma arises due to the trade off between plasticity when changing parameters for new tasks and stability when keeping them the same for prior learned tasks.

The brain is not susceptible to such problems and it is able to adeptly learn many tasks. For example, whilst humans are still quite capable of forgetting, they are able to learn the rules of chess, football, cricket, poker, and many other games sequentially, without forgetting the rules to the others. The mechanisms of how the brain does this are not fully known, though neuroscience experiments examining neuronal activity and plasticity in biological neural networks are providing insights.

One way in which the brain may enable continual learning could be by restricting plasticity of the synaptic connections that are important for a memory once it is formed. Upon the introduction of a new stimulus or task, new excitatory connections are made on dendritic spines in the brain (Bonilla-Quintana & Wörgötter, 2020; Makino & Malinow, 2009). As learning occurs the strength of these connections increases via long-term potentiation, where the response of the synapse for a given stimulus occurs more readily (Alloway, 2001; Yuste & Bonhoeffer, 2001). Furthermore, the connections which were important for prior tasks experience reduced plasticity, thereby protecting these connections from being lost as new tasks are learned (Bliss & Collingridge, 1993; Honkura et al., 2008). This property of biological networks inspired the introduction of Synaptic Intelligence (Zenke et al., 2017) and Elastic Weight Consolidation (EWC) in artificial neural networks (Kirkpatrick et al., 2016), both of which have means of calculating the importance of individual weights to prior tasks. The regularization terms introduced in these methods encourage the network to find solutions that change the most important weights as little as possible.

Another proposed mechanism is the idea that memories are encoded in distinct 'neuronal ensembles', that is to say different sets of neurons are recruited to encode memories in different contexts (Mau et al., 2020). This use of neuronal ensembles may reduce interference between memories and prevent the kind of catastrophic forgetting observed in artificial neural networks (González et al., 2020).

There are three broad categories of methods through which mitigating catastrophic forgetting has been implemented: replay-based methods (Li & Hoiem, 2016; van der Ven & Tolias, 2018; Rebuffi et al., 2016), regularization methods (Kirkpatrick et al., 2016; Zenke et al., 2017; Jung et al., 2016) and architectural methods (Schwarz et al., 2018; Masse et al., 2018; Aljundi et al., 2016). The model we introduce in this paper uses a combination of architectural and regularization methods.

## 1.2 RELEVANT WORK

In 2017, a method was introduced known as Context Dependent Gating (XDG) (Masse et al., 2018) and was demonstrated to successfully alleviate catastrophic forgetting in feedforward and recurrent architectures on image recognition tasks like permuted MNIST and Imagenet. This method works by multiplying the activity of the hidden layers of an artificial neural network by masks or 'gates'. In XDG the gates are binary vectors where a random 20% of the nodes in the hidden layers of the network are allowed to be active for each context and activities for the remaining 80% of units are set to zero. This allows the network to use different sub-networks which have little overlapping representations (though it was allowed, at chance, overlap was roughly 4%). In combination with Elastic weight Consolidation (EWC) (Kirkpatrick et al., 2017), the XDG method achieved strong performance on continual learning benchmarks. For XDG the gates applied for each task were randomly allocated and the context signal used to retrieve them had to be given as an additional input. In many realistic continual learning scenarios a context label is not specified and the allocation of random gates may not be optimal. The method introduced in this paper is an extension of XDG where gates are instead the outputs of trainable networks themselves.

## 1.3 OUR CONTRIBUTION

We introduce learned context dependent gating (LXDG), where the gating of the hidden layers is trainable and is dependent on the input of the tasks. Crucially, an advantage over XDG is that this method does not require any additional label specifying which task is being presented when performing on previously learned tasks. For example, if new data from a previously learned task is presented without the associated context label, it would not be possible for XDG to perform well without finding the context label that was used to train that task. The is because XDG requires a context vector to reproduce the correct gating. Some methods have attempted to address this limitation of XDG, such as the Active Dendrites Network (Iyer et al., 2022). In this case context vectors are still required, but they are derived from a simple statistical analysis of inputs. The LXDG method no longer requires a context vector to be explicitly provided or inferred. The gates for LXDG are learned and dynamically produced using a set of newly introduced regularization terms. The method is able to simultaneously increase task performance and allocate the gates in a single loss function. The random gating in XDG does allow for some level of overlap at chance. However, there is no flexibility for the model to potentially benefit from allowing overlapping learned representations for

similar tasks. This provides a potential advantage for LXDG as the gates are not chosen at random, as they are in XDG. Instead the loss function can encourage the inclusion of units that benefit task performance. Continual learning is demonstrated with this method on the permuted MNIST (Goodfellow et al., 2013) and rotated MNIST benchmark tasks. There are three additional regularization terms required for the method to allocate new sparse gates for new tasks and to recall prior gates for previously learned tasks.

## 2 METHODS

We trained the models on two benchmark tasks, permuted MNIST (Goodfellow et al., 2013) and rotated MNIST. Permuted MNIST is a task in which the pixels of the traditional MNIST images (Lecun et al., 1998) are shuffled to create new tasks. Rotated MNIST tasks were generated by rotating the MNIST images by a random angle, sampled uniformly between 0 and $\pi$. For the training and test set in each task, there are 60,000 and 10,000 images respectively. The model was trained on 28x28 permuted MNIST images across 50 tasks (i.e. 50 unique ways of shuffling or rotating the image), each with 20 epochs. The models are trained with backpropagation with a learning rate of .001 along with batch sizes of 256. All models were trained in Pytorch (Paszke et al., 2019) on NVIDIA's GeForce GTX 1080 Tis.

The architecture of the model is feedforward and shown in figure 1. It includes two fully connected 'forward' hidden layers each followed with a ReLU output activation. Dropout (Srivastava et al., 2014) was included after each layer with a probability of 0.5. There are adjacent gating layers whose output are a multiplier for the output of the 'forward' hidden layers. Each of the forward layers has 2000 units. There is a final output layer for the ten classes. In the gating layers, there are two fully connected layers with 400 units for each of the 'forward' layers. These layers also use a ReLU output activation and are followed by a dropout probability of .5. The output of the gating layers uses a sigmoid output activation. For consistency in the number of trainable parameters, all comparisons are made with an identical network structure, even if the gate regularization terms were not included during training.

### 2.1 REGULARIZATION TERMS

The gating layers require their own regularization terms to perform correctly. We are using the layers to approximate a function that takes the inputs for the task and outputs different sparse gates for each task. Let's call the function we are trying to approximate $g$. The function $g$ should take input data $d$ and output the correct gating $u$ for each context,

$$\underset{i}{\text{layers}}\ g_i(d) = u_i, \tag{1}$$

the resulting vector $u_i$ is used to multiply the activities of each hidden layer $i$.

The method must do three things, allocate a sparse gating for each task, allocate new gating for new tasks and retain stable gates when presented with previously learned tasks. Each of these properties can be expressed as a loss function for the network.

**Sparsity Loss:** Ideally we want $u_i$ to be a sparse binary vector, for gate sparsity we can define a loss function by

$$\mathcal{L}_{sparse} = \sum_i |\frac{g_i(d) \cdot g_i(d)}{N_{\text{hidd}}} - \beta_s|^2, \tag{2}$$

where $\beta_s$ is the sparsity parameter set by default to $\beta_s = 0.2$. Choosing the correct $\beta_s$ is a compromise between optimizing the capacity of the subnetworks and ensuring the overlap of the networks does not cause overwriting of past gates. We kept the $\beta_s = 0.2$ in accordance with the XDG paper. This sparsity loss is only expected to work because the gates are produced with a sigmoid output activation. We find that the artificial neural network tends arrive at a solution where the gate outputs associated with 20% of the nodes have value $\approx 1$ and the rest have value $\approx 0$. Ensuring the outputs of the gates are as close as possible to binary vectors is important for the artificial network to produce something similar to 'neuronal ensembles'.

**Change Loss:** For allocating new gates for new tasks the prior input data for task $t$, $d^t$, can be used to ensure that new gates are as orthogonal as possible from the previously learned gates. These gating

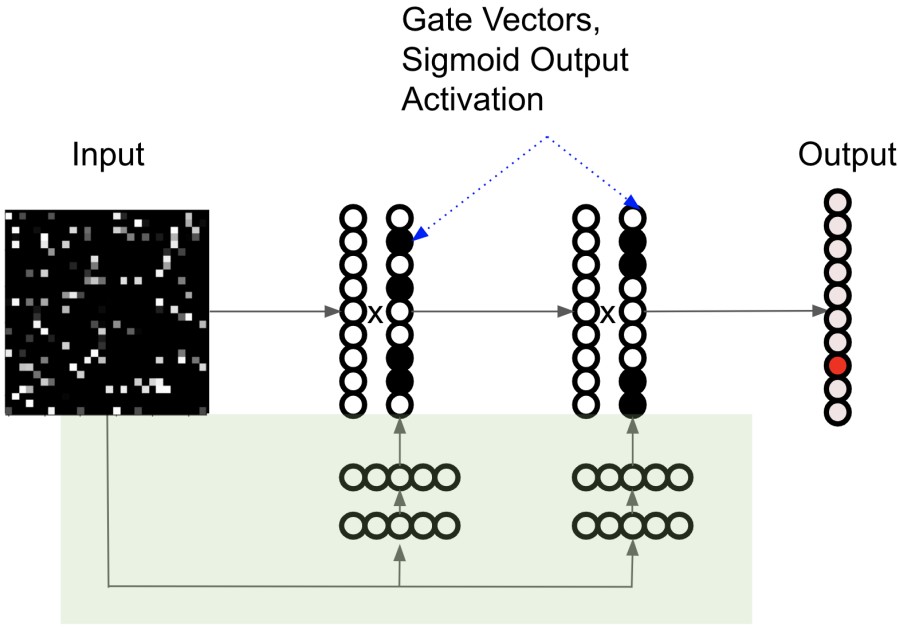

Figure 1: Schematic demonstrating a simplified structure of LXDG. The gating layers are highlighted in green. The black dots indicate gate output values of $\approx 0$. Input is an example permuted MNIST image.

functions $g_i$ take a random sample of data from previously learned tasks $d^t$ to reproduce prior gates. For perfectly stable gating functions the vectors $g_i(d^t)$ are identical to the vectors $g_i^t(d^t)$, where $g_i^t$ are the gating functions after training on prior task $t$. The regularization term that encourages the production of new gates can then be defined as

$$\mathcal{L}_{change}^t = \sum_i |\frac{g_i(d^t) \cdot g_i(d)}{N_{\text{hidd}}} - \beta_c|^2, \tag{3}$$

where each prior task has an associated sample of unlabeled input data $d^t$. To ensure new gates are as orthogonal as possible from all prior gates the change parameter can be set to $\beta_c = 0$. If we were to set $\beta_c > 0$, this would specify a desired overlap with prior learned gates.

**Keep Loss:** For maintaining the stability of gates associated with prior tasks, a final loss can be constructed,

$$\mathcal{L}_{keep}^t = \sum_i |\frac{g_i^t(d^t) \cdot g_i(d^t)}{N_{\text{hidd}}} - \beta_k|^2. \tag{4}$$

The new gating function $g$ should produce the same gates as the prior ones $g^t(d^t)$, when presented with input from prior tasks $d^t$. In this loss function $\beta_k > \beta_s$ to ensure that the new gating function reproduces the vector when presented with inputs from previously learned tasks. By default $\beta_k$ was chosen to be 0.3 as the only requirement for this parameter is that it should be greater than 0.2. This encourages the term $g_i^t(d^t) \cdot g_i(d^t)$ to be maximized up to 0.3. However, as the sparsity should not be greater than 0.2, the term should ideally not reach that maximum. This method does require a batch of input data from prior tasks $d^t$, but it does not require the labels from previously learned tasks.

Lastly, the Elastic Weight Consolidation (EWC) regularization term (Kirkpatrick et al., 2017) is included. The term comes in the form,

$$\mathcal{L}_{EWC}^t = \frac{1}{2}F^t(\theta^t - \theta)^2, \tag{5}$$

for each prior task $t$, where $F$ is the diagonal of the Fisher information matrix, $\theta^t$ are the model parameters at the end of training task $t$ and $\theta$ are the most recent network parameters. The diagonal of the Fisher information matrix can be interpreted as a matrix of parameter 'importance' and the constraint works by preventing important weights from changing too much during training.

The total loss function then becomes,

$$\mathcal{L} = \mathcal{L}_{task} + \lambda_{sparse}\mathcal{L}_{sparse} + \sum_t (\lambda_{change}\mathcal{L}^t_{change} + \lambda_{keep}\mathcal{L}^t_{keep} + \lambda_{EWC}\mathcal{L}^t_{EWC}), \quad (6)$$

where three of the losses are included in a sum over all prior learned tasks $t$. The coefficients for each of the losses are $\lambda_{sparse}$, $\lambda_{change}$, $\lambda_{keep}$, and $\lambda_{EWC}$. Here $\mathcal{L}_{task}$ is the task loss, minimized for best performance. If the method is to be used for reinforcement learning minimizing $\mathcal{L}_{task}$ could maximize future returns instead.

The desired behavior of the gating layers is produced when these regularization terms are simultaneously minimized. The regularization terms allow the network to allocate new sparse gates for new tasks and recall learned gates when presented with inputs from previously seen tasks.

## 2.2 PARAMETER OPTIMIZATION

The coefficients $\lambda_{sparse}$, $\lambda_{change}$, $\lambda_{keep}$ are set to the same values, $\lambda_{gate}$, for simplicity in the optimization. The two parameters in the loss function $\lambda_{EWC}$ and $\lambda_{gate}$ were optimized with a sweep across a range of values. We anticipate that it will be possible with future work to perform a full grid search and to include the different $\lambda$ parameters for each of the introduced regularization terms. However, due to computational constraints, we performed a search by first finding the optimal $\lambda_{EWC}$ parameter and then sweeping across values for $\lambda_{gate}$.

The coefficient for the EWC regularization was first optimized using a scan over values 0.1, 0.2, 0.5, 1, 2, 5, 10, 20, 100, 200, 500, 1000, 2000, 5000 and 10000. The model was trained on the task permuted MNIST and the mean accuracy after 50 tasks was stored for each case. The optimal point was 1000 as shown in figure 2.

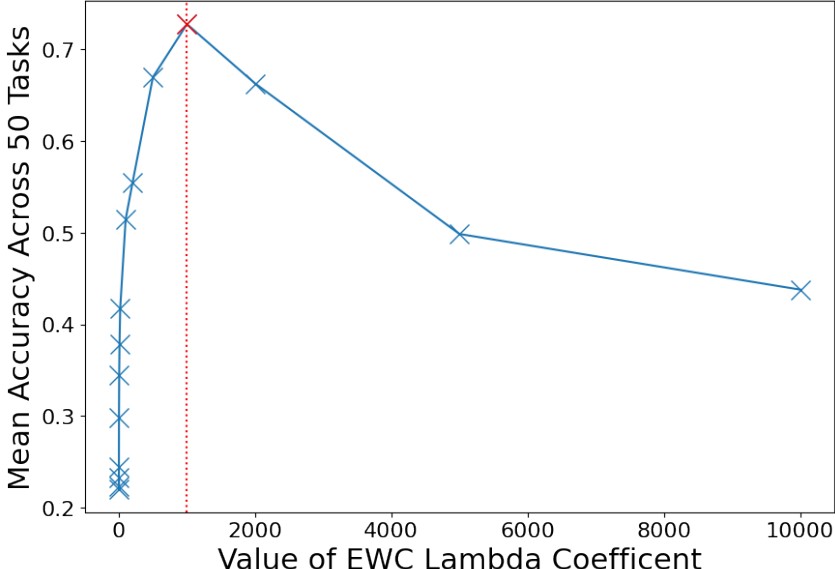

Figure 2: Mean performance across 50 permuted MNIST tasks when training with EWC alone. Scan across 15 values of $\lambda_{EWC}$. The optimal point was found at a value of 1000, shown in red.

With the value of $\lambda_{EWC}$ fixed to 1000 we performed a scan for the gating coefficient, $\lambda_{gate}$. The values used were 0.5, 1, 100, 200, 500 and 1000 for this parameter sweep. The performance increases with $\lambda_{gate}$ to a limit of approximately 95% as shown in figure 3. We chose the optimal value of $\lambda_{gate} = 1000$.

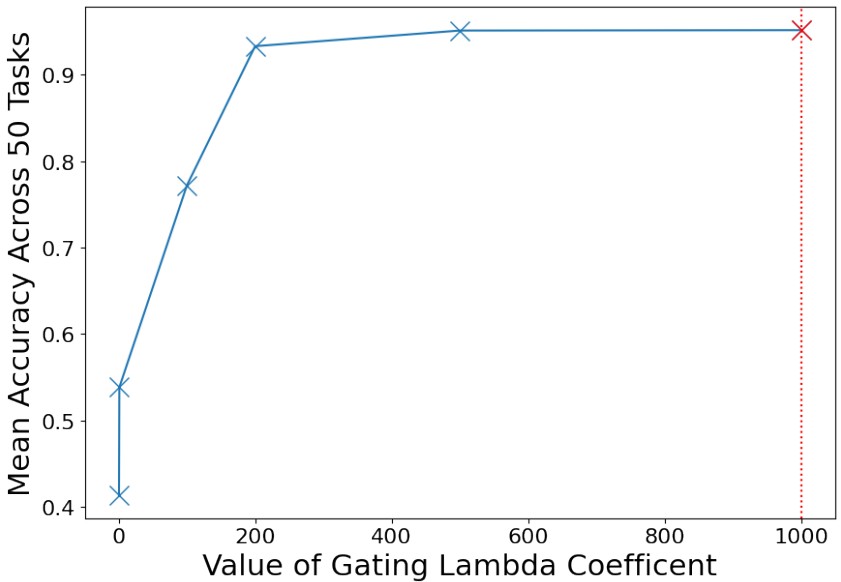

Figure 3: Mean performance across 50 permuted MNIST tasks when training with LXDG + EWC. Scan across 15 values of the $\lambda_{gate}$ Optimal point found at the maximum value of 1000, shown in red.

## 3 RESULTS

### 3.1 PERMUTED MNIST

We trained models on the permuted MNIST (Goodfellow et al., 2013) set of tasks. Each Permuted MNIST task is a unique shuffling of pixels for the MNIST images. As expected, figure 4 shows that the model with no additional regularization terms in the loss catastrophically forgets. As the tasks are presented sequentially the mean accuracy across all learned tasks drops precipitously. For the models including EWC regularization the performance is much better than with no constraints on the loss. The final mean accuracy of EWC across 50 tasks is $72 \pm 1\%(SD)$. The best performing model is XDG+EWC where the gates are random, unlearned and the context is given to the network directly. The performance of LXDG in figure 4 is much closer to XDG+EWC than just EWC alone. The mean accuracy of LXDG+EWC across 50 tasks was $96\%$ whereas for XDG+EWC the mean accuracy was $97\%$. We did not expect LXDG+EWC to outperform XDG+EWC due to the significant advantage of being given the context label directly. There appears to be a small instability for the very first set of tasks trained with LXDG+EWC. However, after 5 tasks the performance remains stable.

To analyze the structure of the gates and how they form as task are introduced we performed principle component analysis (PCA) on the set of learned gates. The mean output of the gates for the final model tested on all 50 tasks was saved. Then PCA was used to obtain the two principle components. The separation of the gates can then be displayed in figure 5. As we would expect unique gates are being found for the tasks. The first gate that was trained is indicated in green for figure 5. It appears to be quite different from the others and may explain the early drop in mean accuracy as the first task does have the lowest accuracy throughout training. The main difference for how this gate was trained when compared to all the others is that at the start of training the network weights are initialized by sampling over a uniform distribution. Further analysis of the first gate shows that it does not choose a binary vector as the solution. See figure 8 in the appendix for the gate output. However, all subsequent gates appear to be allocated correctly with the expected binary vector-like behavior in the outputs. The sparsity in the network weights after training on the first permuted MNIST task may be important for producing the correct gate outputs. For layer 1, the first principle component (PC1) appears to separate the gates by when the associated task was introduced.

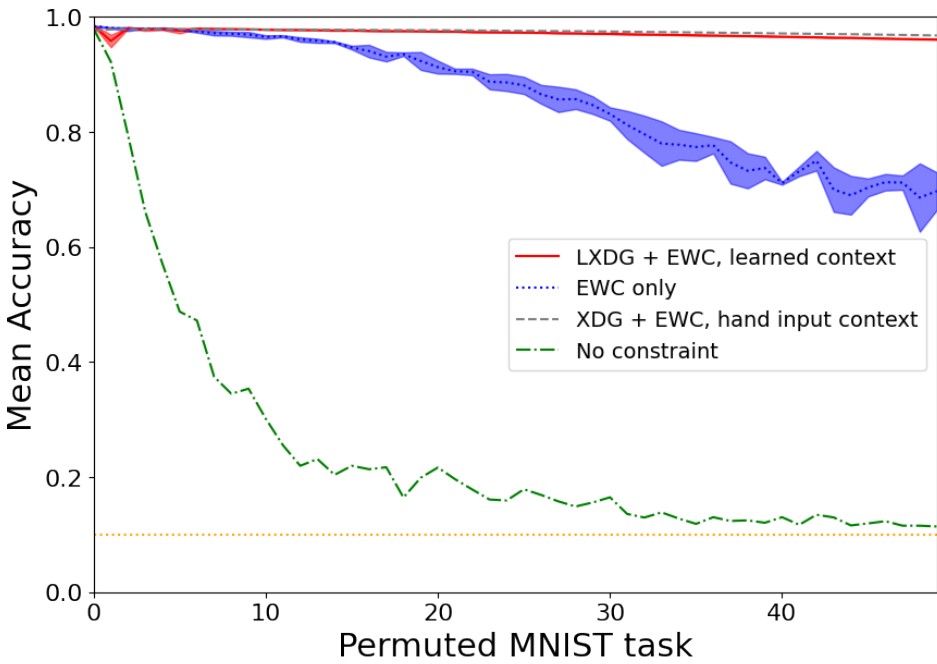

Figure 4: The mean accuracy on 50 permuted MNIST tasks for the LXDG + EWC model (solid red), EWC alone (blue dashed), XDG + EWC (black dashed) and no constraint (green dash-dotted). As the model sees each new task, its performance on the current and all prior tasks is averaged. This is repeated for each task to produce the mean accuracy across tasks. The bands are $\pm 1$ standard deviation in mean accuracy with tasks produced using 3 different random seeds for EWC only and LXDG + EWC. The orange dotted line indicates chance level performance.

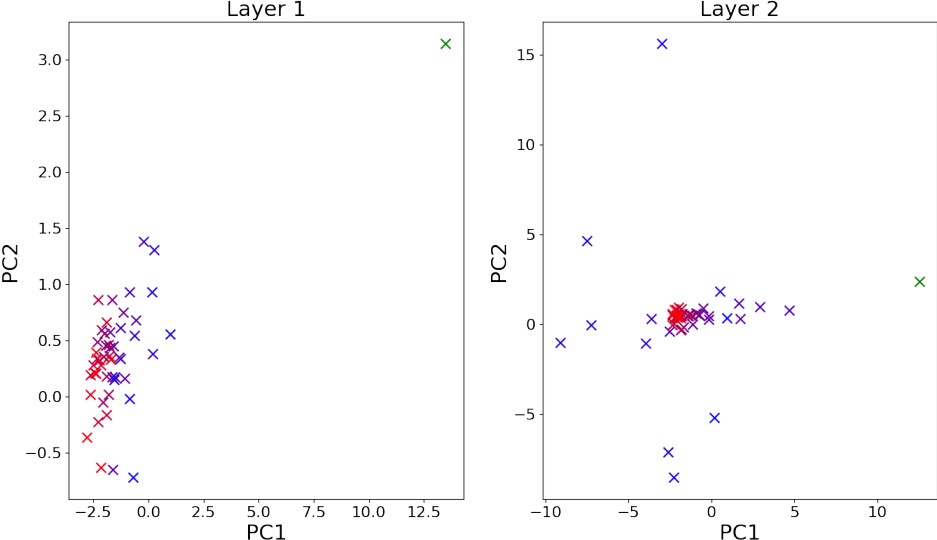

Figure 5: Principle component analysis of recalled gates when the LXDG+EWC model is trained on 50 tasks of permuted MNIST. The gates were analyzed as binary vectors where the gate output was considered 'active' if the mean output of the gate was greater than 0.1. The points are colored in the order that the tasks were presented, from blue to red. The exception is the gate associated with the first task, presented in the color green. The first gate appears to be anomalous and may explain the slight instability in the initial performance observed in figure 4.

## 3.2 ROTATED MNIST

The models were trained on an additional set of tasks, rotated MNIST. Each task was generated by rotating the MNIST images by a random uniformly sampled angle between 0 and $\pi$. The advantage of using this benchmark is that similarity between tasks is easily quantifiable. The performance of the method on rotated MNIST above EWC alone is demonstrated in figure 6. More interestingly for this benchmark, analysis of the produced gates demonstrates that similar tasks produce similar gates. This provides the potential for knowledge of prior learned tasks to transfer to new ones by producing similar gates. In the loss function, the target is still for the gates to be as orthogonal as possible, $\beta_c = 0$. However, to satisfy the other parts of the loss function the network chooses a compromised solution, where the gates for similar tasks are less than orthogonal.

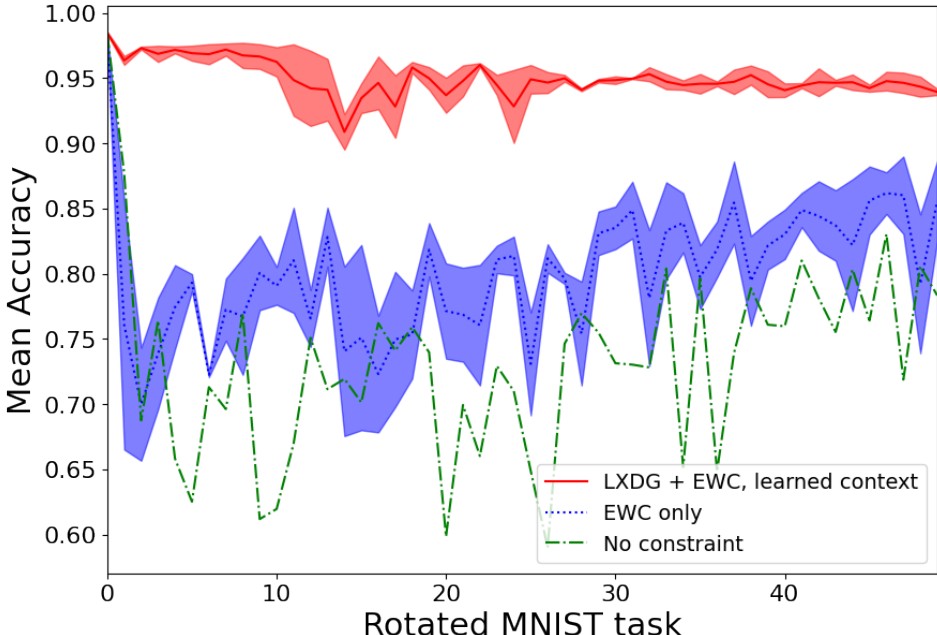

Figure 6: The mean accuracy across 50 tasks of rotational MNIST. The model with LXDG+EWC (red, solid) remains more consistent with higher performance throughout the 50 tasks of rotational MNIST than the model with EWC alone (blue, dotted). The EWC only model and the model with no additional regularization (green, dash-dotted) begins to regain some level of performance later on as it is shown tasks with similar rotations to those previously learned, but does so in a less stable manner than the LXDG+EWC model. The bands are $\pm 1$ standard deviation in mean accuracy with tasks produced using 3 different random seeds for EWC only and LXDG+EWC.

To investigate if the task structure is reflected in the structure of the gates, a principle component analysis was performed on the recalled gate outputs. In figure 6 the gates form a ring structure and the points on the ring are related to the angle that was used to produce the task. Similar tasks with similar rotation angles produce similar gate outputs.

## 4 CONCLUSION

This work demonstrates the ability of LXDG to alleviate catastrophic forgetting for a set of benchmark tasks. Unlike prior work, XDG (Masse et al., 2018), this method of allocating gates to alleviate catastrophic forgetting does not require explicit context labels. Instead, the method is able to perform well on previously learned tasks without specifying which task it was. It achieves this by learning to allocate the correct context gating from the task input alone. The LXDG+EWC method produced good performance on two continual learning benchmarks, permuted MNIST and rotated MNIST.

The rotated MNIST benchmark tasks demonstrates how tasks presented with a similar rotation to a prior learned task will produce a similar gating. This presents the network the opportunity to

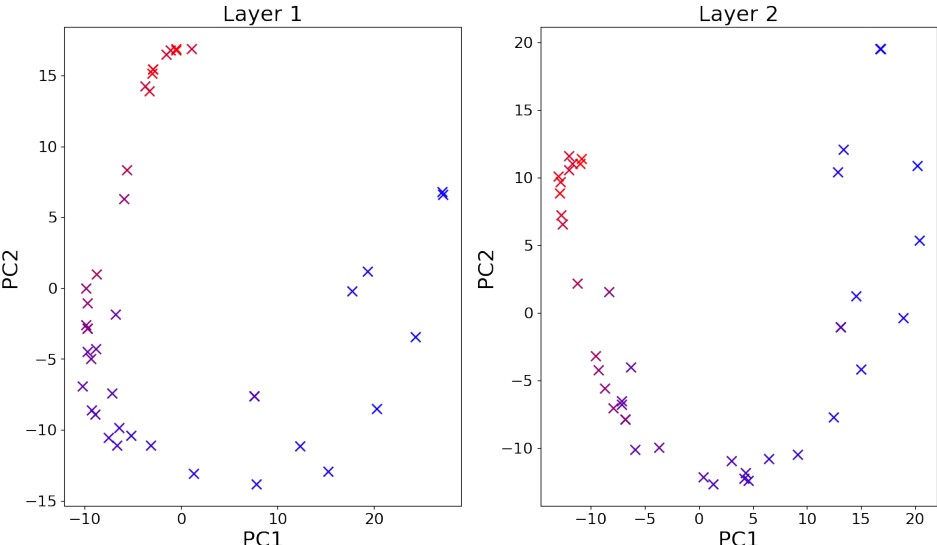

Figure 7: Principle component analysis of the LXDG gate outputs across 50 tasks. The points represent the gate for each task of rotated MNIST. Unlike figure 5, points are color-coded not in the order that the tasks were presented but by the rotation angle of the MNIST images blue (0) to red ($\pi$). Similar angles produce gates close together in the reduced 2 dimensional space. As in figure 5, the gates were analyzed as binary vectors where the gate output was considered 'active' if the mean output of the gate was greater than 0.1.

transfer the information stored in the weights from prior tasks to new similar tasks. This is because the starting place for the model learning a new task will automatically produce gates for tasks similar to those seen previously. We hope that by building on this work, this method can be used to produce transfer learning capabilities in significantly more complicated tasks.

Because the method recalls gates from previously seen input data alone, without the use of the prior task labels, the method can be used in a reinforcement learning (RL) context. Future work should explore the performance of the model on benchmark RL continual learning tasks, such as the Atari 2600 suite of games.

This method was inspired by the biological process of encoding newly acquired memories or task knowledge in neuronal ensembles. For this reason, a future direction of this work would be to use LXDG as a model to compare with how biological neural networks assign and recall neuronal ensembles. With neuroscience experiments performing recordings of neural activity in animals (e.g. monkeys and mice) it will be possible to observe these neuronal ensembles and assess overlap between neuronal ensemble for memories encoded in different contexts. In particular, it is possible to measure the active neurons in an ensemble when a mouse is performing a task in one context vs. performing a similar but different task in another context (Mau et al., 2020). It will be interesting to investigate if biological networks assign ensembles for new contexts in a similar way to how the LXDG method assigns new gates. For example, we are interested in getting LXDG to reproduce the behavior in biological networks where memories encoded closer in time share more overlap in their neuronal ensembles (Cai et al., 2016).

Assigning and recalling neuronal ensembles could be an important step in producing artificial intelligence that can interact with the world reliably, in a multitude of changing contexts. We hope that LXDG can be developed further to help produce flexible artificial neural networks capable of continual learning in challenging environments.

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

## A  APPENDIX

### A.1  PERMUTED MNIST GATE PROPERTIES

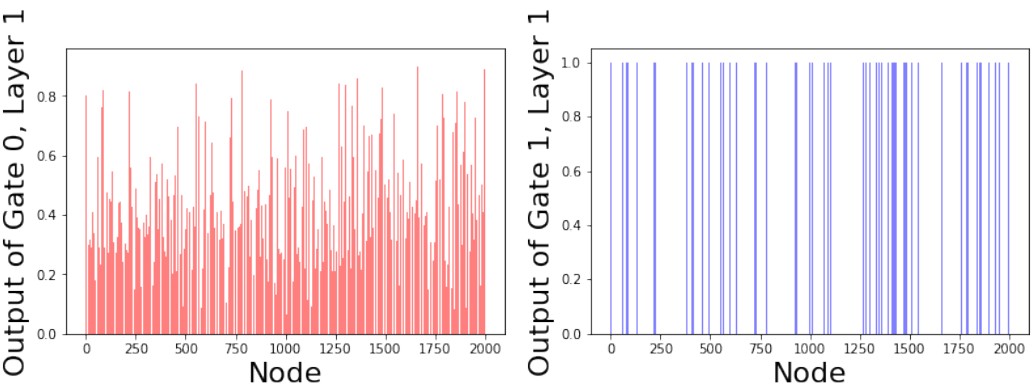

Figure 8: Mean output of the first (left, red) and second (right, blue) gates. Demonstration of the problem with the initial gate. The first gate satisfies the sparsity loss but not by producing a binary vector as the output. However, All other subsequent gates do appear to produce the expected behavior.

### A.2  PERFORMANCE COMPARISON

The recently published model referred to as active dendrites (Iyer et al., 2022) has been added to the performance plot in figure 9. In this model a context vector is still required at train and test time. However, the model includes a method to get the context vector from a simple statistical analysis of the inputs. Therefore, while the method differs substantially from LXDG, which does not use a context vector at all, it is an interesting point of comparison. We have taken the values directly from the paper, indicated by the asterisk in the figure. We would have preferred to reproduce the results ourselves but given time constraints, this was not possible. For this reason we do not want to overstate our confidence in the comparison. The active dendrites model is combined with the similar synaptic intelligence (Zenke et al., 2017) (SI) method instead of EWC. We find SI often marginally outperforms EWC so this should give active dendrites a slight advantage in this comparison. However, with this naive comparison LXDG does outperform the active dendrites method in mean performance at 50 tasks. Conveniently, the model sizes are comparable, active dendrites used two layers of size 2800 and LXDG uses two forward layers of size 2000 and two hidden layers of size 400 for each of the hidden layers. The final mean performance at 50 tasks for LXDG+EWC was 96.0% and active dendrites has a mean performance of 94.4%.

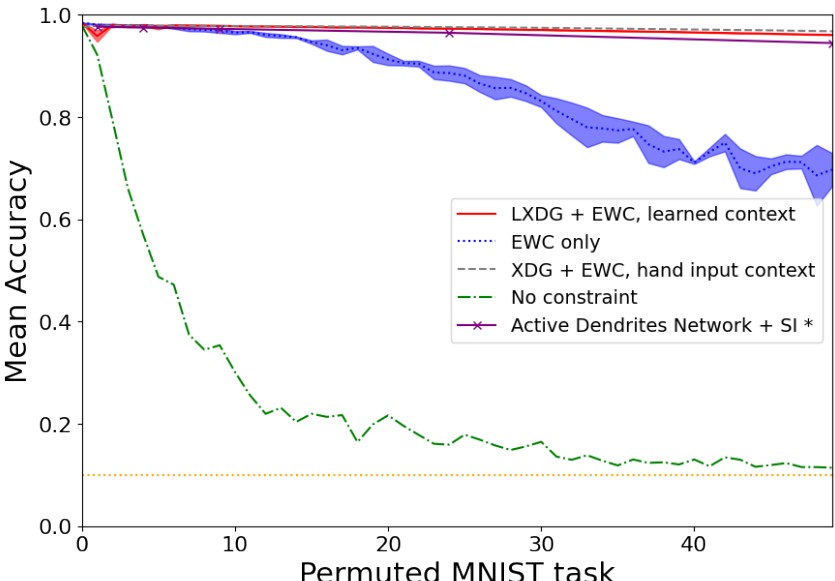

Figure 9: Performance comparison of LXDG + EWC (red solid), EWC only (blue dotted), XDG + EWC (black dashed), no constraint (green dash dot) and active dendrites + SI (purple solid). The asterisk indicates that the values are taken directly from the paper (Iyer et al., 2022) and not reproduced by ourselves.

