# OpenReview forum: "Artificial Neuronal Ensembles with Learned Context Dependent Gating"
_ICLR.cc/2023/Conference — ICLR 2023 poster_

### Official Review · Reviewer_6wb8 · 2022-10-24

**Confidence:** 3
**Clarity, Quality, Novelty And Reproducibility:** Clear enough.
**Correctness:** 4
**Technical Novelty And Significance:** 2
**Empirical Novelty And Significance:** 2
**Recommendation:** 5

**Strength And Weaknesses:**

Strengths: The paper addresses an important problem, and the results represent a significant advance relative to Masses, Grant and Freedman's PNAS paper.  The results, relative to EWC and no constraint are compelling.
Weakness: It is not clear this is the state of the art in continual learning, which is a very active area of research in ML, and has a leaderboard with many different tasks and many algorithms.  This paper does not address any of those tasks. The permuted MNIST and rotated MNIST might be too simple.

**Summary Of The Paper:**

This paper extends the context-dependent gating (XDG) approach to avoid catastrophic forgetting in life-long learning, i.e. learning a new task might overwrite the weights learned for another task. XDG randomly disabled 20% of the neurons in each layer to ensure the creation of distinct neural ensembles for different tasks. This paper proposes to learn these input-dependent gating masks using a neural network. The authors showed that such a network yields superior performance for solving a  continual learning benchmark such as permuted MNIST and rotated MNIST in object recognition.

**Summary Of The Review:**

The paper proposes an extension of the XDG approach to alleviate catastrophic forgetting in life-long learning, with compelling results.  Using a neural network to learn the context dependent gating is new in this context. However, the algorithm has not been rigorously compared with other state of the art algorithms and against more challenging benchmark in  the field of continual learning.  It has promise but the  technical innovation and implications currently are quite limited.

---

> ### Author Response · Authors · 2022-11-12
> **Response to Official Review of Paper5379 by Reviewer 6wb8**
>
> We thank the reviewer for their comments and feedback on our original submission. The reviewer has raised important points about potential limitations of the study and we hope that we have addressed some of the concerns and strengthened the revised manuscript.
>
> > Weakness: It is not clear this is the state of the art in continual learning, which is a very active area of research in ML, and has a leaderboard with many different tasks and many algorithms. This paper does not address any of those tasks. The permuted MNIST and rotated MNIST might be too simple.
>
> This is a good point, and we realize that the manuscript would benefit by addressing this in the revised submission. We accept that demonstrating the LXDG method on two quite simple benchmark tasks, permuted MNIST and rotated MNIST, is a limitation of the study. In fact, this is one of the areas in which we are most actively working in this project, and based on our preliminary findings, we expect to have new results in time for presentation at the conference. Specifically, we are actively working to extend the study further to include more challenging and complicated benchmarks & implement the method in challenging reinforcement learning regimes in a future publication. Although we expect to have new results in time for the ICLR conference, we regret that we are not yet in a position to share new results given the tight timeframe of the rebuttal and the durations required for running revised networks on these new benchmarks.
>
> We agree with the reviewer that more comparisons should be drawn to state-of-the-art work. We consider XDG+EWC to be genuinely state-of-the-art, but with the limitation of requiring the task labels at test time. An appropriate comparison would be with methods recently published in the literature that also attempt to address this limitation. We consider the method referred to as “active dendrites” to be a state-of-the-art continual learning method that is an interesting point of comparison.
>
> We thank the reviewer for raising this important point and we have included the following discussion to section A.2. in the appendix of the paper.
>
> "The recently published model referred to as active dendrites (Iyer et al., 2022) has been added to the performance plot in figure 9. In this model a context vector is still required at train and test time. However, the model includes a method to get the context vector from a simple statistical analysis of the inputs. Therefore, while the method differs substantially from LXDG, which does not use a context vector at all, it is an interesting point of comparison. We have taken the values directly from the paper, indicated by the asterisk in the figure. We would have preferred to reproduce the results ourselves but given time constraints of the rebuttal this was not possible. For this reason we do not want to overstate our confidence in the comparison. The active dendrites model is combined with the similar synaptic intelligence (Zenke et al., 2017) (SI) method instead of EWC. We find SI often marginally outperforms EWC so this should give active dendrites a slight advantage in this comparison. However, with this naive comparison LXDG+EWC does outperform the active dendrites+SI method in mean performance at 50 tasks. Conveniently, the model sizes are comparable, active dendrites used two layers of size 2800 and LXDG uses two forward layers of size 2000 and two hidden layers of size 400 for each of the hidden layers. The final mean performance at 50 tasks for LXDG+EWC was 96.0% and active dendrites has a mean performance of 94.0%"
>
> Figure 9 demonstrates the comparison.
>
> Once again, the reviewer's comments have helped strengthen the revised submission substantially. Please let us know if there is anything further that we can address in the remaining rebuttal time.

---

### Official Review · Reviewer_LEqE · 2022-10-24

**Confidence:** 3
**Correctness:** 4
**Technical Novelty And Significance:** 4
**Empirical Novelty And Significance:** 3
**Recommendation:** 8

**Clarity, Quality, Novelty And Reproducibility:**

The paper is clearly written and of expected quality. While learning of gating functions is an obvious thought, someone needed to properly explore and implement it. The design of appropriate regularization terms for the gating function is novel to me. Since this work is mostly experimental in nature, reproducibility is hard for me to judge.

**Strength And Weaknesses:**

Strengths:
The paper systematically compares the new method to alternatives and provides clear evidence of superiority.

Weaknesses:
I see no significant weakness.


**Summary Of The Paper:**

Context-Dependent Gating (XDG), proposed by Masse, Grant, and Freedman, showed promise in dealing with catastrophic forgetting. In that work, gating is randomly chosen before training. This work follows up with the obvious next question: how to learn gating for different tasks. Authors show Learned Context Dependent Gating (LXDG), combined with Elastic Weight Consolidation (EWC), outperforms existing methods on permuted MNIST and rotated MNIST datasets.

**Summary Of The Review:**

This work is a natural extension of XDG, well executed.

---

> ### Author Response · Authors · 2022-11-12
> **Response to Official Review of Paper5379 by Reviewer LEqE**
>
> We thank the reviewer for their time and the encouraging evaluation of our work.
>
> >Since this work is mostly experimental in nature, reproducibility is hard for me to judge.
>
> After leaving double-blind review we intend on releasing a codebase on github, which will allow reproducibility to be directly assessed by the community.

---

### Official Review · Reviewer_wV5E · 2022-10-25

**Confidence:** 3
**Correctness:** 4
**Technical Novelty And Significance:** 2
**Empirical Novelty And Significance:** 3
**Recommendation:** 5

**Clarity, Quality, Novelty And Reproducibility:**

The text is well-written and the experiments are well-documented making the study reproducible.

**Strength And Weaknesses:**

This work contributes to the field of continual learning by introducing a context-aware mechanism of gating the network neurons while learning.

Among the strengths of the work are:

-A nice and simple model that learns to identify the tasks to for the gates (I am surprised that it hasn’t been done yet!)

-High performance, matching that of the previous approaches, but with no need to supply task labels

-I enjoyed the analysis which shows what similar tasks (i.e. MNIST data rotated to small angles) lead to similar gates.

With that said, I believe that additional analysis could showcase the abilities of this model and thus strengthen this contribution. It looks like the content-aware gating which generates similar gates for similar tasks should be more parsimonious than the existing approaches and thus should be able to pack more tasks in the same neural network. Can the Authors please comment on that proposition and/or run such an experiment? It shouldn’t take long.


**Summary Of The Paper:**

This paper addresses the problem of catastrophic forgetting in deep neural networks by introducing a complementary network for gating the neurons. This second network is trained based on the same input to produce task-aware gates which are sparse and regularized to be similar within each task while orthogonal between the tasks. The authors test their architecture on permuted- and rotated-MNIST tasks and show that their model, with no supplied labels, exhibits a performance close to that of the existing models which required task labels.

**Summary Of The Review:**

The paper adds a simple yet powerful tweak to the existing continual learning approaches allowing to omit task labels at the evaluation time. I believe that the model can be further explored to assess its other properties such as a potential increase in the number of tasks learnable with a single network. Such properties – if shown – may strengthen the paper.

---

> ### Author Response · Authors · 2022-11-12
> **Response to Official Review of Paper5379 by Reviewer wV5E**
>
> We appreciate the reviewer's helpful feedback on our original submission. We hope to address the interesting concerns raised in order to further improve the manuscript.
>
> > I believe that additional analysis could showcase the abilities of this model and thus strengthen this contribution. It looks like the content-aware gating which generates similar gates for similar tasks should be more parsimonious than the existing approaches and thus should be able to pack more tasks in the same neural network. Can the Authors please comment on that proposition and/or run such an experiment? It shouldn’t take long.
>
> We thank the reviewer for bringing up this important point about the research. While the XDG+EWC method has the limitation of requiring the task labels at testing time, LXDG has a potential advantage in that it is content-aware. The high mean performance across tasks demonstrates a kind of task ‘packing’ for LXDG. Unfortunately, comparing with XDG+EWC is unlikely to demonstrate the content-aware advantage at 50 tasks. The advantage of explicitly being given the task labels at testing time does seem to outweigh the benefits to efficiently packing tasks in LXDG.
>
> One comparison we have added and discussed in the appendix is with a model known as “active dendrites” (Iyer et al., 2022). This model still requires a context vector but it is calculated from a simple statistical analysis of the inputs. We find in a naive comparison that LXDG outperforms active dendrites at 50 tasks of permuted MNIST. The values for the active dendrites model are taken directly from the paper. We have added section A.2 in the appendix to discuss this comparison in more detail. We suggest it could be that having a more content-aware approach and not requiring a context vector is the reason that the number of performative tasks ‘packed-in’ to LXDG is greater.
>
> We actively investigating ways to address this point further and we thank the reviewer for raising it.

---

> > ### Comment · Reviewer_wV5E · 2022-11-16
> > **Re: response**
> >
> > Thank you for your response.
> >
> > It would be still great if you consider generating more than 50 permuted-MNIST tasks using the same approach or, perhaps, use another benchmark (I gather that you are currently working on additional benchmarks anyway, as per your response to another Reviewer) to see how many tasks could be learned by your model, compared to the baselines.

---

> > > ### Author Response · Authors · 2022-11-19
> > > **Response to Comment From Reviewer wV5E**
> > >
> > > To address this comment we recently began running LXDG+EWC with the intent of going above 50 tasks of permuted MNIST. However, we left it too late and it has only reached a little above 50 tasks as of writing this response. We don’t anticipate that it will be finished with substantially more tasks in time to update the manuscript before the Nov 18th deadline. This is a priority to improve the research. We are able to extend beyond 50 tasks of permuted MNIST and we also intend to test the model on other benchmarks before the ICLR 2023 conference date.
> > >
> > > Additionally, we may soon have the opportunity to resolve some of our computational limitations by upgrading from our GTX 1080 Ti GPUs, which should help us make comparisons at much larger numbers of tasks.
> > >
> > > We thank the reviewer again and regret that we couldn't prepare a more satisfying response in time for the rebuttal deadline.

---

> > > > ### Comment · Reviewer_wV5E · 2022-11-23
> > > > **Re**
> > > >
> > > > Thanks for your response and cooperation. Please keep us posted even after the Nov 18 deadline.

---

### Official Review · Reviewer_jDUA · 2022-10-25

**Confidence:** 3
**Correctness:** 4
**Technical Novelty And Significance:** 4
**Empirical Novelty And Significance:** 4
**Recommendation:** 8

**Clarity, Quality, Novelty And Reproducibility:**

The paper is very well-written and easy to read.

The work is well-scoped and novel in an appropriate, measured way.

I am not versed in the literature of catastrophic forgetting, so I cannot confirm originality.

I did not see mention of an open codebase.

**Strength And Weaknesses:**


(strengths)

The background context is clear.  The paper is well-scoped, and the paper delivers incremental but significant improvement on prior work.

The findings are convincing.

The work has significant value.

Miscellaneous comments:

Punctuation errors: 2 commas near top of page 2, period in first line of page 5.

Please number equations.

Re lambda_gate: better might be to give different lambdas in the equations, then in the first paragraph of 2.2 note that they were all set equal. Alternately (and more standard I think), withhold the lambdas from the individual loss equations, and include them in the collected Loss eqn at top of pg 5.

I did not get the last paragraph of 3.1. Is this a meaningful observation? ie, is the network in the paper similar enough to bio networks to draw any conclusions?

In section 3.2: Did the value of beta_c (in L_change) change since the tasks were so closely related?

Fig 5 and 7: please specify the task type (permuted or rotated) in the caption title.


**Summary Of The Paper:**

The paper develops a method of adding gates to hidden units in order to protect their weights from modification during training on new tasks. The goal is to prevent catastrophic forgetting. The method, LXDG, extends a prior method XDG, by (among other things) removing the need for a task label.

**Summary Of The Review:**

I loved this paper. Well-written, interesting, well-scoped, with convincing and valuable findings.

---

> ### Author Response · Authors · 2022-11-12
> **Response to Official Review of Paper5379 by Reviewer jDUA**
>
> We thank the reviewer for this encouraging assessment of our work. Your comments have helped us strengthen the submission.
>
> > Punctuation errors: 2 commas near top of page 2, period in first line of page 5.
>
> We believe we have addressed the period on page five which should have been a comma. We have revised the paragraph at the top of page 2 for clarity. We are happy to make further changes.
>
> > Please number equations.
>
> Thank you for this suggestion, we have done this.
>
> > Re lambda_gate: better might be to give different lambdas in the equations, then in the first paragraph of 2.2 note that they were all set equal. Alternately (and more standard I think), withhold the lambdas from the individual loss equations, and include them in the collected Loss eqn at top of pg 5.
>
> We have modified the paper to use the latter approach and withheld the lambdas from the individual loss equations. They are then included in the full loss equation.
>
> > I did not get the last paragraph of 3.1. Is this a meaningful observation? ie, is the network in the paper similar enough to bio networks to draw any conclusions?
>
> We have been quite interested in making direct comparisons to biology as a future direction of this work. However, we agree that this paragraph was too speculative. We believe that other readers will have the same reaction and so we have taken this part out of section 3.1. Instead we have added a little more discussion in section 4 about where we want to go in the future when making comparisons to biology.
>
> While LXDG is very disanalogous to biology in many ways. It could still be used as a simplified model for studying memory building via neuronal allocation. One way we have been interested in using LXDG is to model how memories learned closer in time are encoded in more overlapping ensembles. We are interested in developing the model to reproduce the behavior of real ensemble dynamics in the mouse hippocampus. While this is speculative work at the moment, it is something we are actively investigating for a follow up publication.
>
> > In section 3.2: Did the value of beta_c (in L_change) change since the tasks were so closely related?
>
> This is a good question, which we have clarified in the revised manuscript. We did not change the value of beta_c, the target is still orthogonal. Though to satisfy the other parts of the loss the network settles on a compromised solution where the gates for similar tasks are less than orthogonal. This question does highlight a future direction we wish to take the work in, we are very interested in exploring how the method performs when setting the target overlap, beta_c, to values greater than zero.
>
> > Fig 5 and 7: please specify the task type (permuted or rotated) in the caption title.
>
> Thank you, we have corrected this.
>
> Through addressing these comments we believe the paper has been substantially improved. Please let us know if there is anything further that we can address in the remaining rebuttal time.

---

> ### Comment · Reviewer_jDUA · 2022-11-23
> **Response to authors' comments and reviewers**
>
> Thank you to the reviewers for interesting comments, and to the authors for their detailed responses to each.
> I am very positive about this paper, because the contribution is so clear, well-framed, and potentially useful. I wish to comment from that perspective:
>
> I was happy to read that the code will be posted to github (cf reviewer LEqE). This adds much value.
>
> Reviewer wE5e raised an insightful line of inquiry re task packing, which the authors appear to address well. Can this response be injected into the paper, ideally in the main body or perhaps in an Appendix?
>
> Reviewers 6wb8 and wE5v rightfully point out that experiments with other tasks would be valuable additions. Since choosing tasks for experiments is always a gamble (so many options, each person has favorites) and since it is hard to do careful experiments on short notice, I respectfully propose the following plan:
>
> 1. Reviewers 6wb8 and wE5v provide short lists of interesting tasks (optional - as convenient).
>
> 2. We accept the paper and trust the authors to run and analyse one or more of these experiments.
>
> 3. The authors reward this trust with the new findings (whatever they are), at the conference and on github.
>
> Thanks much.

---

> > ### Comment · Reviewer_wV5E · 2022-11-23
> > **Re**
> >
> > Thanks for initiating the discussion.
> >
> > Would you mind elaborating on what makes this paper a clear-accept-recommendation material? Unless I'm missing something here, both reviews suggesting this grade do not provide many details when it comes to the basis of such an assessment.
> >
> > Please don't get me wrong, I like the paper and acknowledge its advances over XDG, the baseline model. I believe that these results need to be published; the question is whether it is going to be here or elsewhere.
> >
> > This paper offers a way of learning the task identities without having explicit labels, but so do the other works (e.g. Wortsman et al, Supermasks in Superposition, NeurIPS 2020). The model here learns similar gates for similar tasks, but the other continual-learning models also exploit similarities in task structure (e.g. Ke et al, Continual Learning of a Mixed Sequence of Similar and Dissimilar Tasks, NeurIPS 2020). The proposed model may be efficient in the number of learned tasks, but the competition is high in that regard too (e.g., again, Wortsman et al), and the clear-cut advantage is yet to be shown.
> >
> > I do not (and did not) request additional benchmarks, because I don't think that an additional permuted/rotated/splitted mnist/fashion-mnist/cifar/mini-imagenet, conventionally used in continual learning, would provide new insights about the properties of the proposed algorithm. Instead, I was looking for some useful properties/applications which the proposed model may enable, unlike the existing ones.
> >
> > With that said, I'm open to opinions and clarifications about what I could have potentially overlooked.

---

### Decision · Program_Chairs · 2023-01-20

**Decision:**

Accept: poster

**Justification For Why Not Higher Score:**

This paper is sound and provides a clear contribution, but it is not ground-breaking and its novelty is real, but not huge. It should not be a spotlight in my estimation, then.

**Justification For Why Not Lower Score:**

Given that the critiques were largely about whether there was enough empirical data, and the scores were above 6, I am inclined to not reject. However, this could be revisited if the SAC feels the paper is indeed to limited a contribution.

**Metareview: Summary, Strengths And Weaknesses:**

This paper proposes a solution to catastrophic forgetting that uses learned gating mechanisms to keep separate neuron ensembles for tasks in different contexts. This allows them to engage in continual learning on relatively simple tasks with better results than other approaches, such as elastic weight consolidation. Importantly, the proposed system does not require labels for the context, which makes it potentially more applicable to a wider range of tasks (though this is not actually demonstrated explicitly).

There was a bit of a split in the reviews, with two reviewers viewing the paper very positively, and two viewing it as being just below the acceptance threshold. However, I did not consider this a borderline case as the reviews were actually quite consistent, there was just different emphasis placed by different reviewers. The principal concerns highlighted by the more negative reviewers was that there were relatively limited demonstrations and discussion of the capabilities of the model relative to other approaches to continual learning. The authors attempted to address these concerns through some more data and more discussion of limitations, but did not have time to present a great deal more data. Nonetheless, it was the AC's judgement that the authors made an honest attempt to address the concerns, and that all the reviewers agreed that the paper was perfectly technically sound and potentially of use in some applications, and so, given the final average score of 6.5, an accept decision was reached.

**Note From Pc:**

if the above contains the word "oral" or "spotlight" please see: "oral" presentation means -> notable-top-5% and "spotlight" means -> notable-top-25%. As stated in our emails, we are disassociating presentation type from AC recommendations

**Summary Of Ac-Reviewer Meeting:**

N/A